Diagnostic value of albumin/fibrinogen ratio and C-reactive protein/albumin/globulin ratio for periprosthetic joint infection: a retrospective study

Ji Wei
Liu Zemiao zemiao1976@163.com
Lin Tao
Department of Joint Surgery, Qilu Hospital of Shandong University (Qingdao) , Qingdao , China
Zhan Cheng
Electronic publication date: 2023 Dec 15
Publication date: 2023
Volume: 11
Electronic Location ID: e16662
Received 2023 Sep 26; Accepted 2023 Nov 21
Copyright: © 2023 Ji et al.
Copyright year: 2023
Copyright holder: Ji et al.
License: This is an open access article distributed under the terms of the Creative Commons Attribution License, which permits unrestricted use, distribution, reproduction and adaptation in any medium and for any purpose provided that it is properly attributed. For attribution, the original author(s), title, publication source (PeerJ) and either DOI or URL of the article must be cited.
License URL: https://creativecommons.org/licenses/by/4.0/

Keywords: Periprosthetic joint infection, Albumin to fibrinogen ratio, C-reactive protein/albumin/globulin ratio, Diagnostic value

Funding: The authors received no funding for this work.

==============================
Background

The study aims to explore diagnostic value of albumin/fibrinogen ratio (AFR) and C-reactive protein (CRP)/albumin (ALB)/globulin (GLO) ratio (CAGR) for periprosthetic joint infection (PJI).

Methods

A retrospective analysis was conducted on clinical data collected from 190 patients who had joint replacement surgery in Qilu Hospital of Shandong University (Qingdao), from January 2017 to December 2022. Based on the occurrence of PJI after surgery, patients were divided as an infection group (10 cases) and non-infection group (180 cases). Diagnostic indicators were analyzed, univariate and multivariate logistic regression analyses were further performed to identify factors related to PJI. Sensitivity and specificity of AFR and CAGR, both individually and in combination, were calculated using ROC curves, and their diagnostic performance was compared based on the area under the curve (AUC).

Results

Levels of CRP, ESR, FIB, GLO, and CAGR were significantly higher in the infection group than in non-infection group (P < 0.05). Levels of ALB and AFR were significantly lower in infection group (P < 0.05). Multivariate logistic regression analysis reviewed that CRP (OR = 3.324), ESR (OR = 2.118), FIB (OR = 3.142), ALB (OR = 0.449), GLO (OR = 1.985), AFR (OR = 0.587), and CAGR (OR = 2.469) were factors influencing PJI (P < 0.05). The AUC for AFR and CAGR in diagnosing PJI were 0.739 and 0.780, while AUC for their combined detection was 0.858.

Conclusion

Abnormal levels of AFR and CAGR are associated with PJI, and their combined use has certain diagnostic value for PJI.

Introduction

Periprosthetic joint infection (PJI) is recognized as one of the most serious complications post artificial joint replacement. According to statistics, incidence of PJI is about 1.2%, among which incidence of PJI in knee joint is about 1.4%, and that of hip joint PJI occurs in approximately 0.9% (Huotari, Peltola & Jämsen, 2015). The management of this complication is extremely difficult. The American Musculoskeletal Infection Society (MSIS) in 2014 included serum C-reactive protein (CRP) and erythrocyte sedimentation rate (ESR) as secondary diagnostic criteria for PJI (Parvizi, Gehrke & International Consensus Group on Periprosthetic Joint Infection, 2014); thus, diagnostic accuracy of PJI was improved. However, there is currently no test index that can claim 100% sensitivity and specificity for PJI diagnosis. Clinically, debridement, antibiotics, irrigation, and retention (DAIR) with retained prosthesis, one or two-stage revision surgeries (Li et al., 2020; Bjerke-Kroll et al., 2014) are commonly used for PJI patients. The choice of treatment strategy is believed to be related to the timing of PJI occurrence. Therefore, current research focuses on discovering simple, rapid, and cost-effective biomarkers for diagnosing PJI.

Fibrinogen (fibrinogen, FIB) is a coagulation-related serological marker, studies have shown that there is an important correlation between the coagulation cascade and inflammatory mechanisms (Davalos & Akassoglou, 2012). Fibrinogen (FIB) and albumin (Alb) are involved in both inflammatory responses and coagulation pathways. FIB can mediate the synthesis of pro-inflammatory factors through the nuclear factor transcription pathway, participate in the occurrence and development of inflammation, and promote platelet aggregation by binding to platelet surface glycoprotein receptors. Alb is a negative inflammatory protein and negatively correlated with red blood cell aggregation. Previous studies have found that FIB and albumin (ALB)/globulin (GLO) ratio (A/G) have certain value in the diagnosis of PJI (Hu, Fu & Tang, 2020; Saleh et al., 2019; Jiao et al., 2022). AFR refers to the ratio of serum albumin to plasma fibrinogen, which is a hematological parameter. Generally speaking, an AFR of around 1.0 is the normal range. If the AFR is high, it indicates liver function damage, malnutrition, inflammation, and other conditions in the body. If the AFR is low, it indicates that there may be vascular endothelial damage, kidney function damage, and other conditions in the body. According to recent literature reports, albumin/fibrinogen ratio (albumin/fibrinogen ratio, AFR), as a new serological marker, has shown excellent clinical diagnostic value in inflammatory and malignant tumor diseases, which has attracted attention (Sun, An & Lv, 2020; Jiang & Lei, 2022; Zhang et al., 2021; Maimaiti et al., 2021). In addition, recent studies have demonstrated that the C-reactive protein/(albumin/globulin) ratio (CRP/ALB/GLO ratio, CAGR) has high value in PJI diagnosis (Choe et al., 2023). C-reactive protein (CRP) is a biomarker of acute inflammation, while albumin (ALB) and globulin (GLB) are two important proteins in the human body. ALB and GLB have different physiological functions in the human body, and their ratio can reflect the nutritional status and immune function in the body. Few studies have demonstrated the value of the AFR in the PJI diagnosis, and there are few studies on the CAGR. Therefore, this study aims to investigate the value of AFR and CAGR in PJI diagnosis by retrospectively analyzing the clinical data collected from 190 patients who had joint replacement from January 2020 to December 2022.

Material and Methods

Research data

Clinical data of 190 patients who had joint replacement in Qilu Hospital of Shandong University (Qingdao) from January 2020 to December 2022 were retrospectively analyzed. Based on whether PJI occurred after operation, patients were further divided into an infection group of 10 cases as well as a non-infection group of 180 cases. Diagnostic criteria of PJI: Based on 2014 edition of MSIS diagnostic criteria for PJI (Parvizi, Gehrke & International Consensus Group on Periprosthetic Joint Infection, 2014), a diagnosis of PJI is made when patients has one of the two major criteria or three of the five minor criteria. Main diagnostic criteria refers to sinus tract communicating with prosthesis; the same bacteria were cultured twice. Secondary diagnostic criteria: increased ESR and increased CRP; increased white blood cell count in synovial fluid or ++ on leukocyte esterase (LE) test paper; increased ratio of neutrophils in synovial fluid; positive single bacterial culture; periprosthetic Histopathological analysis was positive. All samples obtained in this study were approved by the ethics committee of the Qilu Hospital of Shandong University (Qingdao) and abided by the ethical guidelines of the Declaration of Helsinki, and ethics committee agreed to waive informed consent.

Inclusion criteria: (1) patients who underwent joint replacement in the hospital for the first time, all of which were unilateral joint replacement; (2) complete clinical data.

Exclusion criteria: (1) chronic inflammatory diseases, infectious diseases, or immune system abnormalities existed before surgery; (2) diseases such as active arthritis, osteomyelitis, rheumatoid arthritis, and severe osteoporosis; (3) patients with severe organ dysfunction or tumors.

Collection of clinical data

We reviewed the medical records and collected the patients’ information, including age, gender, body mass index, affected side as well as other information.

Lab tests methods

On the morning after admission, 2 ml of venous blood was collected in EDTA-K2 anticoagulant tubes for CRP measurement using the immunoturbidimetric method on a specialized protein analyzer PA-990 (Sysmex, Kobe, Japan). ESR was detected using the kinetic method on Alifax TEST1 analyzer (Alifax, Polverara, Italy). Fasting venous blood samples of 3 ml were collected in coagulation-promoting tubes, centrifuged to separate serum, and ALB levels were measured using the dye-binding method on an ARCHITECT C18000 automated biochemical analyzer (Abbott, Chicago, IL, USA). GLO levels were calculated from the ALB levels. A total of 2 ml venous blood was placed in tubes with citrate anticoagulant, centrifuged to separate plasma. FIB levels were detected using the Clauss method on the automated coagulation analyzer CS-5100 (Sysmex, Kobe, Japan). CRP, ESR, FIB, ALB, and GLO results data of patients who met the inclusion criteria were collected and recorded from the hospital’s electronic medical record system. Value of AFR (ALB/FIB) and CAGR (CRP/(ALB/GLO)) were further calculated.

Statistical analysis

SPSS 21.0 (SPSS Inc., Chicago, IL, USA) was used for analysis, and measurement data conforming to normal distribution was represented as X¯±S. Comparison between the two groups was performed by way of group t test. Count data was expressed as number of cases or rate. The χ2 test is adopted for the comparison of the two groups. The variables that had statistical significance in univariate analysis are further included in multivariate analysis. Multivariate analysis adopts logistic regression model. ROC curve is adopted to evaluate influence of related factors on PJI for diagnostic value. P < 0.05 was recognized as statistically significant.

Results

Comparison of basic information between the two groups

No significant differences were revealed in age gender, body mass index, and side between the two groups (P > 0.05), as illustrated in Table 1.

Table 1 Comparison of basic information of patients in the two groups.

	Infection group (n = 10)	Non-infection group (n = 180)	t/χ2 value	P value	
Age (year)	63.98 ± 12.26	63.10 ± 12.75	0.213	0.832	
Gender (case)					
Male	8	98	2.508	0.113	
Female	2	82			
Body mass index (kg/m2)	24.38 ± 2.76	24.89 ± 2.86	0.550	0.583	
Side (case)					
Left	7	86	1.872	0.171	
Right	3	94			

Comparison of two groups of laboratory indicators

The levels of CRP, ESR, FIB, GLO, and CAGR in the infection group were remarkably higher than those in non-infection group (P < 0.05), while levels of ALB and AFR were remarkably lower than those in non-infection group (P < 0.05), as illustrated in Table 2.

Table 2 Comparison of laboratory indicators between the two groups.

	Infection group (n = 10)	Non-infection group (n = 180)	t value	P value	
CRP (mg/L)	42.65 ± 10.54	8.95 ± 2.65	29.938	<0.001	
ESR (mm/h)	54.98 ± 15.09	22.45 ± 5.61	15.662	<0.001	
FIB (g/L)	5.87 ± 1.57	3.42 ± 0.87	8.234	<0.001	
ALB (g/L)	37.08 ± 7.43	40.79 ± 4.16	2.612	0.009	
GLO (g/L)	32.76 ± 6.32	26.05 ± 4.52	5.652	<0.001	
AFR	7.94 ± 3.13	12.96 ± 3.36	4.613	<0.001	
CAGR	39.84 ± 14.92	6.68 ± 1.63	28.107	<0.001	

Multiple linear regression analysis of influencing factors of PJI

Variables that were statistically significant in univariate analysis were used as the independent variables, and PJI was categorized as dependent variable (No = 0, Yes = 1) for multivariate logistic regression analysis. Results showed that CRP(OR(Odds Ratio) = 3.324), ESR(OR = 2.118), FIB(OR = 3.142), ALB(OR = 0.449), GLO (OR = 1.985), AFR(OR = 0.587), CAGR(OR = 2.469) are the influencing factors of PJI (P < 0.05), as illustrated in Table 3.

Table 3 Multiple linear regression analysis of influencing factors of PJI.

Factor	Beta value	SE value	Ward value	OR value	95% CI	P value	
CRP	1.201	0.376	10.205	3.324	[1.590–6.945]	0.000	
ESR	0.732	0.269	8.132	2.118	[1.265–3.582]	0.004	
FIB	1.264	0.623	4.275	3.142	[1.076–11.653]	0.035	
ALB	−0.800	0.121	43.792	0.449	[0.354–0.569]	0.000	
GLO	0.683	0.332	4.182	1.985	[1.036–3.821]	0.010	
AFR	−0.532	0.143	13.878	0.587	[0.443–0.776]	0.000	
CAGR	0.903	0.336	7.235	2.469	[1.277–4.770]	0.000	

The diagnostic value of AFR and CAGR alone and combined detection for PJI

The ROC curves demonstrated that for PJI diagnosis, AFR and CAGR had AUC of 0.739, 0.780. Used in combination, the AUC value for PJI diagnosis increased to 0.8581, as illustrated in Table 4 and Figs. 1–3.

Table 4 Diagnostic value of AFR and CAGR alone and combined detection for PJI.

	AUC	95% CI	Best cut-off value	Sensitivity (%)	Specificity (%)	
AFR	0.739	[0.596–0.881]	9.03	88.93	62.54	
CAGR	0.780	[0.653–0.909]	15.35	91.43	64.65	
Joint detection	0.858	[0.756–0.960]	\	92.68	81.25	

Figure 1 ROC curve for AFR in PJI diagnosis.

Figure 2 ROC curve for CAGR in PJI diagnosis.

Figure 3 ROC curve for combined detection in PJI diagnosis.

Discussion

PJI is recognized as a severe complication post joint replacement surgery, and many patients require multiple surgical interventions due to uncontrolled infection or recurrent infection after treatment. The use of antibiotics during treatment can lead to various complications, including organ failure and even death (Huang et al., 2019; Wang et al., 2022). The overall poor treatment outcomes in patients with PJI are closely related to the difficulty in diagnosing PJI. In the early stages of PJI, patients may only experience localized pain without systemic infection symptoms such as chills and high fever, making accurate diagnosis challenging. Therefore, many researchers have been dedicated to finding more reliable and efficient diagnostic markers. β-defensin-3 and Toll-like receptors (TLRs) have been recently identified as biomarkers with high diagnostic performance for PJI. However, their clinical application is limited due to the requirement for specialized equipment and complex procedures (Galliera et al., 2014; Liu et al., 2014). On the other hand, AFR and CAGR can be easily calculated based on commonly used laboratory tests. They are simple, fast, and cost-effective diagnostic indicators, making early diagnosis of PJI possible in primary healthcare settings.

AFR combines two important biomarkers: ALB and FIB. ALB reflects both nutritional status and inflammatory state. When the immune system is activated, the albumin level decreases. FIB, as coagulation factor I, is found to be one of the indicators associated with blood clotting and inflammation (Wang et al., 2021; Jennewein et al., 2011). FIB not only promotes platelet aggregation and increases blood viscosity by binding to platelet fibrinogen receptors but also increases the permeability of fibrous caps in atherosclerotic plaques, leading to thinning and vulnerability of the plaques. When the plaques rupture, procoagulant factors within the lipid core enter the bloodstream, resulting in thrombus formation. FIB can also upregulate pro-inflammatory factors synthesis, including interleukin-1, tumor necrosis factor, thus mediating occurrence and development of inflammatory reactions. Studies have found that during severe infections, ALB decreases while FIB increases due to the cascade reactions of coagulation and inflammation (Davalos & Akassoglou, 2012). The results of this study illusrated that AFR had prominent diagnostic value for PJI (AUC = 0.739). In arthroplasty, Maimaiti et al. (2021) found that, low AFR values were effective biomarkers for postoperative malnutrition in the study of 466 patients received revision arthroplasty, demonstrating that, AFR had high predictive value (AUC = 0.721) for predicting acute PJI after revision surgery. Jiang & Lei (2022) found that low AFR values were independent risk factors for postoperative delirium in the study of 336 patients received total hip arthroplasty.

CRP is found to be an acute-phase protein synthesized by liver, in response to stimulation by factors related to stress or microbial infections. CRP significantly increases when the body undergoes physical or chemical damage or microbial infections. CRP is widely used as an infection diagnostic marker in clinical practice. Wu et al. (2022) found that the postoperative CRP level is a related predictive indicator of the efficacy of knee joint prosthetic infection replacement surgery, and the CRP level of patients in the infection cured group after replacement surgery is lower than that in the infection failed group. ALB is a major component of human serum proteins. It is negatively correlated with the inflammatory state of the body (Farrugia, 2010). During infection, the level of albumin often decreases. Hypoalbuminemia is an important manifestation of malnutrition and can be used to diagnose malnutrition. Additionally, low albumin levels can serve as diagnostic indicators of infection and inflammation in the body. GLO is another important component of serum proteins. It includes immunoglobulins, various glycoproteins, complement components, and plasma ceruloplasmin. When the body is in an inflammatory state, the level of globulin increases significantly. The GAGR combines the levels of albumin and globulin and can better reflect the level of inflammation in the body. It is a good diagnostic indicator of inflammation. Wu et al. (2022) found that CAGR performed well in diagnosing PJI based on ROC analysis (AUC: 0.902 and 95% CI [0.845–0.943]; optimal cutoff value: 5.08). Choe et al. (2023) also demonstrated the high diagnostic value of CAGR in PJI. Sensitivity and specificity were 96.0% and 92.0%, while AUC value was 0.980 with a cutoff value of 3.1. Compared to this study, with a cutoff value of 15.35, sensitivity and specificity were 91.43%, 64.65%, while AUC value was 0.780. The findings of this study revealed that the combined use of AFR and CAGR for PJI diagnosis had an AUC value of 0.858, indicating that the combined detection of AFR and CAGR could reflect the degree of peri-prosthetic inflammation effectively, improving the diagnostic value of PJI. AFR and CAGR are relatively new and valuable indicators, and there is limited research on their diagnostic value in PJI. Therefore, further exploration by more researchers is needed.

Conclusions

Abnormal levels of AFR and CAGR in PJI patients have certain diagnostic value. However, the current study has a few limitations: (1) It is a non-prospective study with selection bias, and the influence of confounding factors cannot be completely excluded; (2) the study did not consider the impact of other comorbidities on AFR and CAGR; (3) the number of eligible cases that met the inclusion criteria was limited, which reduced the reliability of the data to some extent. These limitations and shortcomings point to the direction for further research on the significance of AFR and CAGR in PJI diagnosis.

Supplemental Information

Supplemental Information 1 Raw Data.

Click here for additional data file.

Additional Information and Declarations

Competing Interests

Author Contributions

Human Ethics

Data Availability

The authors declare that they have no competing interests.

Wei Ji conceived and designed the experiments, analyzed the data, authored or reviewed drafts of the article, and approved the final draft.

Zemiao Liu performed the experiments, analyzed the data, prepared figures and/or tables, and approved the final draft.

Tao Lin conceived and designed the experiments, performed the experiments, analyzed the data, prepared figures and/or tables, authored or reviewed drafts of the article, and approved the final draft.

The following information was supplied relating to ethical approvals (i.e., approving body and any reference numbers):

All samples obtained in this study were approved by the ethics committee of the Qilu Hospital of Shandong University (Qingdao) and abided by the ethical guidelines of the Declaration of Helsinki.

The following information was supplied regarding data availability:

The raw data are available in the Supplemental File.

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
