# Peer review of "Diagnostic value of albumin/fibrinogen ratio and C-reactive protein/albumin/globulin ratio for periprosthetic joint infection: a retrospective study"

_PeerJ, doi:10.7717/peerj.16662_

## Round 0.1 · original submission · Major Revisions

Please revise the manuscript as the reviewers suggested.

Reviewer 1 ·

Basic reporting

The manuscript explores the diagnostic efficacy of AFR and CAGR in prosthetic joint infection (PJI). The authors critically demonstrate that abnormal levels of AFR and CAGR play a significant role in diagnosing PJI. The manuscript is well-written in clear, technically correct English. The introduction and background adequately frame the work within the broader field of knowledge around PJI diagnosis, illustrating a distinct need for further investigation of AFR and CAGR. The study's primary strength lies in its commitment to filling this knowledge gap. However, the study's methodology and technical standards could have been portrayed with more detail and rigour, notably a more in-depth explanation of the analytical methods used and the replication potential of these methods. The underlying data does seem accurate, and necessary control measures are apparent, though the statistical analysis could be more detailed, especially with regards to ensuring robustness. Overall, the study is a significant addition to the field of PJI diagnostics, but it falls short in certain areas that could undermine the validity of its findings.

Experimental design

1. The authors should consider referencing their laboratory methods, especially if they are less common or novel.
2. Analytical methods should be clearer. Please consider providing a few more details about the multivariate analysis.
3. The authors need to provide the numerical values of CRP, ESR, FIB, ALB, and GLO as evidence for their claims.

Validity of the findings

1. Specify how "AFR" and "CAGR" were calculated, as the reader may not be familiar with these measures.
2. The authors should clarify what they mean by "specialized equipment and complex procedures." Additionally, reference studies showing that β-defensin-3 and TLRs are indeed effective PJI biomarkers.
3. Expand on why other comorbidities might have influenced the results.
4. Explain the significance of the AUC values of 0.739 and 0.780.

Additional comments

1. Define the terms AFR and CAGR before introducing their importance.
2. The authors might consider evaluating all abbreviations in complete terms when first introduced for clarity.
3. Some sentences are lengthy and could be more concise. Besides, several sentences have grammatical errors that impact readability.

Reviewer 2 ·

Basic reporting

The study provides insightful analysis about the potential use of AFR and CAGR as diagnostic indicators for PJI. The English used in expressing the content is adequately technical and clear; however, there are instances where it lacks precision. The article launches off from a clear and logical introduction, linking this investigation to the broader picture of PJI effectively. I am persuaded by the authors' identification of the knowledge gap that few studies exist on the diagnostic value of AFR and CAGR for PJI. However, bridging this gap could have been handled better. The technical standards used fall short of providing a true benchmark, the level of detail in the technical aspect of the research, particularly the multivariate analysis, is lax and thus, makes it hard for the study to be duplicated accurately. While the data seems to be sound, available, and controlled, further details on the statistical analysis used for robustness would provide the necessary backup for these findings. Overall, although the study presents a compelling argument for using AFR and CAGR to diagnose PJI, it requires further enhancement in its technical details and methodology for a more robust presentation.

Experimental design

1) The authors should explain the connection between ALB and malnutrition. A reference would be useful here, and without it, the argument lacks scientific rigor.
2) Provide evidence for the claim that CRP is an acute phase protein synthesized by the liver. Explain why it occurs, or provide a citation.
3) It is not clear what the comparison between the previous studies and the current study contributes to the reader's understanding of the results. This needs clarification.

Validity of the findings

1) Paragraph 7: The authors need to clarify the term "influence" in the sentence – do they mean a correlation or causation?
2) The authors should provide all the AUC values for the AFR and CAGR. Also, clearly indicate which Figures show these data.
3) Provide references for the claim that FIB is associated with blood clotting and inflammation.
4) Clarify how AGR can "better reflect the level of inflammation." Provide evidence for this assertion.
5) Provide evidence to support the claim that the combined detection of AFR and CAGR can effectively reflect peri-prosthetic inflammation.

Additional comments

1) Please differentiate GLB from GLO. Is it a typo?
2) Sometimes the term 'CAGR' is used, sometimes 'AGR'. Be consistent in your terminology.
3) Paragraph 18: Explain what is meant by 'selection bias.'
4) The abbreviation OR has not been expanded at its first use. Always expand abbreviations at their first use, even if they are common.

Reviewer 3 ·

Basic reporting

.

Experimental design

.

Validity of the findings

.

Additional comments

Thank you for the opportunity reviewing this work.

The authors report results of Albumin/Fibrinogen Ratio and C-reactive protein
/Albumin/Globulin Ratio in the PJI diagnosis. This retrospective study showed that these
two ratios presented reasonable results in sensitivity and specificity for diagnosing PJI.
PJI diagnosis is one of the most challenging scenarios for and orthopedic surgeon. And
every effort looking for new tools are welcome. Nonetheless, we, as surgeons, should be
looking for universal tools or exams that could outperform classic parameters as CRP
and ESR. In general, I personally think these two ratios were not as convenient as CRP
and ESR since these two ratios were not presented in routine lab examinations.
Moreover, a ratio is a composite of two parameters each of which may increase or
decrease. Thus, the use of ratio's cloud what is happening.
However, although I personally see no advantage to look over these two ratios basing on
this study which doesn't mean the study is not valid.

---

## Round 0.2 · accepted · Accept

This manuscript can now be accepted.

Reviewer 2 ·

Basic reporting

no comment

Experimental design

no comment

Validity of the findings

no comment